# On the genus *Sabidius* Strelzov, 1973 (Annelida: Paraonidae), with a redescription of the type species and the description of a new species

Natália Ranauro[1]*, Rômulo Barroso[2], Paulo Cesar Paiva[3], João Miguel de Matos Nogueira[1]

**1** Laboratório de Poliquetologia (LaPol), Departamento de Zoologia, Instituto de Biociências, Universidade de São Paulo, São Paulo, SP, Brazil, **2** Instituto de Biologia, Universidade Federal da Bahia, Salvador, BA, Brazil, **3** Laboratório de Polychaeta, Departamento de Zoologia, Universidade Federal do Rio de Janeiro, Rio de Janeiro, RJ, Brazil

☯ These authors contributed equally to this work.

\* nataliaranauro@gmail.com

## Abstract

Two species of *Sabidius* Strelzov, 1973 were collected during a benthic survey, from 10–3,000 m deep, in Espírito Santo Basin, off southeastern Brazilian coast. Those species are *S. cornatus* (Hartman, 1965) and *S. antennatus* **sp. nov.** The genus *Sabidius* was monotypic until the present study, in which a new species is proposed, together with a redescription of the type species, with expansion of the geographic distribution of the genus and of the type species. The diagnostic feature of the genus is the morphology of prostomium, with crested anterior margin. The main feature that differentiates the two species within the genus is the presence/absence of a median antenna, which is present in the new species described in the present study and absent in *Sabidius cornatus*.

## Introduction

Members of the family Paraonidae Cerruti, 1909 are among the most abundant polychaetes in benthic communities, with high species richness [1]. They are present in all oceans, dominant in soft bottom communities, particularly in the deep sea [2]. They are motile deposit feeders, apparently feeding on both surface and subsurface sediments [3].

Currently, there are approximately 140 species of paraonids, distributed in eight genera: *Aparaonis* Hartman, 1965 [4] (1 sp.), *Aricidea* Webster, 1879 [5] (~75 spp.), *Cirrophorus* Ehlers, 1908 [6] (7 spp.), *Levinsenia* Mesnil, 1897 [7] (22 spp.), *Paradoneis* Hartman, 1965 [4] (20 spp.), *Paraonella* Strelzov, 1973 [8] (7 spp.), *Paraonis* Cerruti, 1909 [9] (6 spp.), and *Sabidius* Strelzov, 1973 [8] (1 sp.) [10]. However, this number is likely to be underestimated, since the family has a long history of taxonomic problems and several new species have been described recently, due to the utilization of finer meshes and the recent increase of projects focusing on the diversity of deep-sea environments [11], which are still undersampled.

**Data Availability Statement:** The data are held in three polychaete collection: Natural History Museum of Los Angeles, Museu Nacional, Universidade Federal do Rio de Janeiro and Museu de Zoologia, Universidade de São Paulo. Accesion numbers are available in the material examined in the manuscript.

**Funding:** NR received a MsC fellowship from CAPES (Coordenação de Aperfeiçoamento de Pessoal de Nível Superior) (proc. 1586471) and currently receives a PhD fellowship from CNPq (proc. 140725/2018-0) CAPES (Coordenação de Aperfeiçoamento de Pessoal de Nível Superior) https://www.capes.gov.br/ CNPq (Conselho Nacional de Desenvolvimento Científico e Tecnológico) http://www.cnpq.br/ PCP receives productivity grant from CNPq (Conselho Nacional de Desenvolvimento Científico e Tecnológico) (proc. 304321/2017-6) and FAPERJ (Fundação Carlos Chagas Filho de Amparo à Pesquisa do Estado do Rio de Janeiro) (proc. E-26/202607/ 2019) CNPq (Conselho Nacional de Desenvolvimento Científico e Tecnológico) http://www.cnpq.br/ FAPERJ (Fundação Carlos Chagas Filho de Amparo à Pesquisa do Estado do Rio de Janeiro) http://www.faperj.br/ JMMN receives a productivity grant from CNPq (Conselho Nacional de Desenvolvimento Científico e Tecnológico), level 2 (proc. 309599/2018-0). CNPq (Conselho Nacional de Desenvolvimento Científico e Tecnológico) http://www.cnpq.br/ We are thankful to CENPES/PETROBRAS for providing the material used for the present study. http://www.petrobras.com.br/pt/nossas-atividades/tecnologia-e-inovacao/

**Competing interests:** No authors have competing interests.

In spite of the number of new species described recently [12–16], for both monotypic genera, *Aparaonis* and *Sabidius*, no additional species has been described since the original descriptions of *Aparaonis abyssalis* Hartman, 1965 [4] and *Sabidius cornatus* (Hartman, 1965) [4]. Although the former species is known from a single specimen [10], members of the latter have been found in several other surveys [2, 10, 17, 18].

*Sabidius cornatus* was originally described as a member of *Paraonis*, but later Strelzov [8] erected the genus *Sabidius* for this species. Morphologically, these animals are similar to members of *Levinsenia*, in the absence of neuropodial lobes and median antenna, but differ from those in having the anterior margin of prostomium with distinctively thick, tri-lobed cuticle.

*Sabidius cornatus* was described based on material from off New England, from slope and abyssal bottoms, between 400–2,900 m deep [4, 17]. Posteriorly, members of this species were found among material from the western Pacific [2], along the U.S Atlantic slope, from the Canadian boundary to off the Carolinas [18], and from slope depths in the Gulf of Mexico, off Louisiana [10].

Regarding the Brazilian coast, few taxonomic studies on the Paraonidae have been conducted [19–21]. Previous studies reported the presence of five genera and 30 species of Paraonidae from Brazilian shoreline, most of them from shallower waters [22].

The aim of this study is to describe a new species of *Sabidius* and re-describe *S. cornatus*, including amendments on the geographic and bathymetric ranges of the latter, based on type material and specimens collected from off southeastern Brazil, state of Espírito Santo, 19˚3'S 37˚44'W–21˚10'S 38˚28'W. The present study is the first record for the genus *Sabidius* in the Southern Atlantic Ocean.

## Material and methods

The specimens were collected from off the state of Espírito Santo, 19˚3'S 37˚44'W–21˚10'S 38˚ 28'W, southwestern Atlantic, during a survey conducted and coordinated by CENPES/PET-ROBRAS (Brazilian Energy Company), under the scope of the AMBES project ("Caracterização Ambiental Marinha da Bacia do Espírito Santo e Porção Norte da Bacia de Campos"). No permits were required for this work. In Brazil before any energy company start to explore a Basin for oil they are obligated to make an environmental characterization of the basin they will explore. The material used in this work came from a project as part of the environmental characterization of the Espírito Santo Basin. As CENPES/PETROBRAS are a Brazilian government company there are no needs of permits to do the sampling.

The oceanographic surveys were completed on board of the oceanographic vessels Seward Johnson and Luke Thomas, from December 2010 to July 2013. Sampling was performed at four different areas: mouth of Rio Doce; continental shelf; slope and canyons (Fig 1). A total of 20 sampling sites were chosen in the area in front of the mouth of Rio Doce, from 10–51 m deep. Sampling was performed along seven transects from south to north. Each transect was composed of ten sampling stations, four of which were in the continental shelf, at 25, 40, 50, and 150 m deep, respectively, and six along the slope, at 400, 1000, 1300, 1900, 2300 and 3000 m deep, comprising 70 sampling sites. Sampling campaigns were also made at Cânion Watu Norte (CANWN) and Cânion Rio Doce (CAND), along depth gradient, 150, 400, 1000, and 1300 m deep. Samples were collected triplicately, by means of a 294 l van Veen grab or a 125 l Box-Corer. Each of the four areas was sampled twice, once in summer and again in winter.

Identifications were based on specific morphological characters. Specimens were examined under stereomicroscope, compound light microscope and scanning electron microscope (SEM). For the SEM, specimens were dehydrated in a series of progressively increasing concentrations of ethanol (70–100%), then critical point dried (LEICA EM CPD300), coated with ~35 nm of gold and examined and photographed at Laboratório de Microscopia Eletrônica

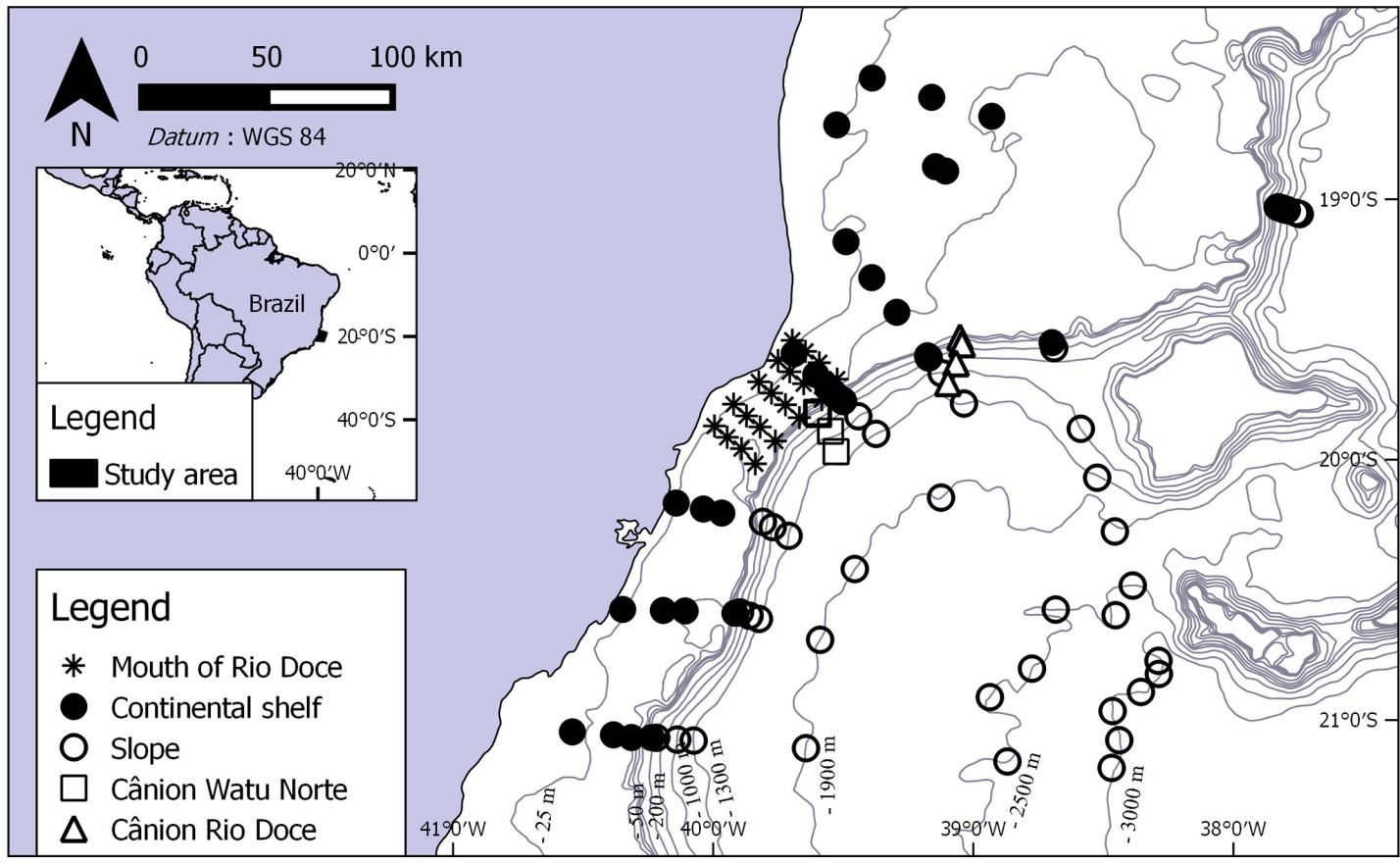

**Fig 1. Study area.** Sampling stations of project AMBES at four different areas: Mouth of Rio Doce, continental shelf, slope and canyons.

(NEMA/PUC-Rio), Laboratório de Microscopia Eletrônica do Museu Nacional (UFRJ) and Laboratório de Microscopia Eletrônica, Instituto de Biociências, Universidade de São Paulo (IB–USP).

Description was based on the holotype and the variation observed on the paratypes is provided inside parentheses. Only complete specimens were measured in length; width was measured at chaetiger 4.

Specimens are deposited in the Museu Nacional, Universidade Federal do Rio de Janeiro (MNRJ) and Museu de Zoologia, Universidade de São Paulo (MZUSP).

## Nomenclatural acts

The electronic edition of this article conforms to the requirements of the amended International Code of Zoological Nomenclature, and hence the new names contained herein are available under that Code, from the electronic edition of this article. This published work and the nomenclatural acts it contains have been registered in ZooBank, the online registration system for the ICZN. The ZooBank LSIDs (Life Science Identifiers) can be resolved and the associated information viewed through any standard web browser by appending the LSID to the prefix "http://zoobank.org/". The LSID for this publication is: urn:lsid:zoobank.org:pub: 12021D56-0819-46A9-BC58-DAAFA840D837. The electronic edition of this work was published in a journal with an ISSN, and has been archived and is available from the following digital repositories: PubMed Central, LOCKSS.

## Results

### Systematics

**Family Paraonidae Cerruti, 1909.** *Genus Sabidius Strelzov, 1973.* Type species. *Paraonis cornatus* Hartman, 1965.

**Diagnosis (emended after Blake, 2016 [10]).** Body long and slender with numerous segments, easily broken. Prostomium with distinctive thick, tri-lobed cuticle on anterior margin; *median antenna present or absent*; ciliated bands, terminal sensory organ and cheek organs all absent; curved nuchal organs on posterior margin of prostomium, often fused to each other, extending across dorsum. Three prebranchial segments present; branchiae restricted to few segments. Neuropodia without podial lobes; notopodial lobes may be present on posterior body notopodia. Chaetae include slender notopodial capillaries, curved modifed spines on postbranchial neuropodia, of which at least 1–2 per fascicle with long arista. *Pygidium with three digitiform anal cirri.*

**Remarks.** The generic diagnosis is emended above, to allow for the presence of a median antenna, as observed among members of *S. antennatus* **n. sp.** (see below), which is absent in *S. cornatus*.

We consider the morphology of prostomium, with crested anterior margin, the numbers of prebranchial and branchial chaetigers, and the cirriform morphology of branchiae, as the diagnostic characters for members of *Sabidius*. Therefore, the presence of a medial antenna is variable within the genus, as also occurs among members of three other genera of paraonids, *Aricidea* [23], *Cirrophorus* and *Paradoneis* [15]. The presence of a median antenna is most likely a homoplastic character in paraonids, given that it has been used to either separate genera or species within a genus [16].

***Sabidius cornatus* (Hartman, 1965).** (Figs 2–4).

*Paraonis cornatus* Hartman 1965: 140–141; Hartman & Fauchald 1971: 102.

*Sabidius cornatus*. Strelzov 1979: 179–180.

**Material examined.** *Type series.* North Atlantic, USA, New England, Continental slope, upper end of canyon just west of Atlantis Canyon, anchor dredge, R/V ATLANTIS Sta. Slope 4, coll. Sanders, H., Woods Hole Oceanographic Institution, 2. Holotype (LACM-AHF POLY 655): 35 chaetigers incomplete, 5.4 mm long, 0.2 mm wide, coll. 28 August 1962, 39˚56'30"N 70˚39'54"W, 400 m. Paratypes: 52 specs (LACM-AHF POLY 656), all incomplete, coll. 28 August 1962, 39˚56'30"N 70˚39'54"W, 400 m.

**Brazilian material.** 2 specs. (MZUSP–3898), coll. 30 Dec 2011, 21˚04'09.61"S 40˚13'7.38"W, 396 m; 3 specs. (MZUSP–3899), coll. 9 Jan 2012, 20˚14'19.45"S 39˚48'36.67"W, 416 m; 2 specs. (MZUSP–3900), coll. 14 Jan 2012, 19˚46'34.99"S 39˚30'04.65"W, 402 m; 3 specs. (MZUSP–3901), coll. 14 Dec 2011, 19˚36'26.24"S 39˚10'17.35"W, 352 m; 1 spec. (MZUSP–3902), coll. 9 Dec 2011, 19˚31'51.66"S 39˚3'4.04"W, 140 m; 2 specs. (MZUSP–3903), coll. 19 Jun 2013, 20˚14'17.95"S 39˚48'34.35"W, 438 m; 2 specs. (MZUSP–3904), coll. 30 Jun 2013, 19˚34'20.47"S 38˚41'19.8"W, 438 m.

**Diagnosis.** Prostomium without median antenna, nuchal organs as curved slits on posterior margin, associated with two areas of reddish-brown pigmentation, often fused and extending across dorsum. Three prebranchial segments; branchiae short, narrow. Notopodia and neuropodia both lacking podial lobes. Notochaetae as slender capillaries; neurochaetae including enlarged, curved, modified spines, of which at least 1–2 per fascicle with long arista, on postbranchial segments. Pygidium an expanded lobe, anal cirri not observed [10].

**Description.** All specimens incomplete, with 35 chaetigers (16–44). Body long and slender, fragile, easily broken. Preserved specimens white–yellowish, without pigmentation patterns other than prostomial spots. Prostomium longer than wide, with distinctive thick, tri-

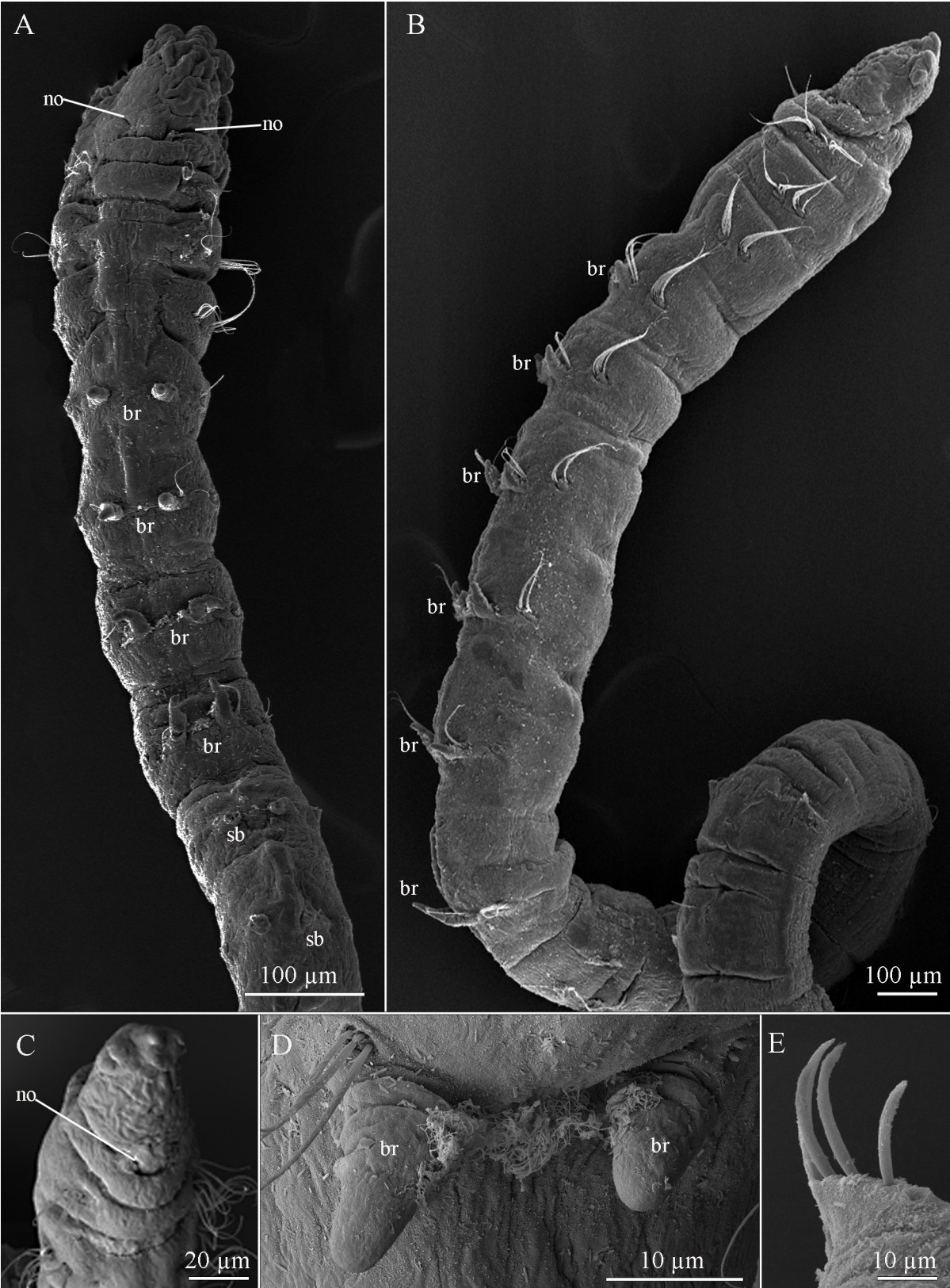

**Fig 2.** *Sabidius cornatus* (A–B). Anterior chaetigers, dorsal and lateral views, respectively; (C). Prostomium, frontal-dorsal view; (D). Pair of branchiae, anterior view; (E). Curved modified spines. br = branchiae; sb = scar of branchiae; no = nuchal organ.

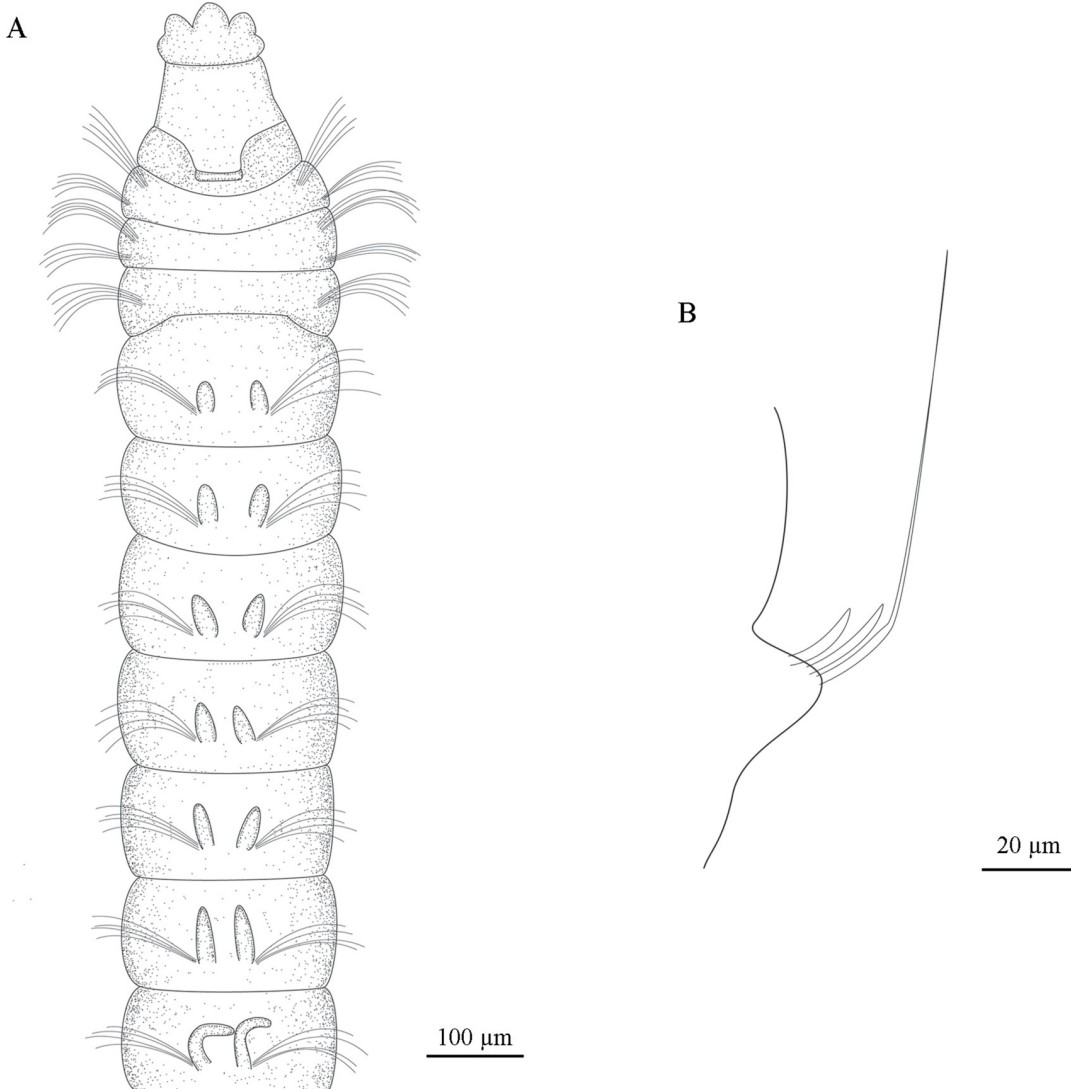

**Fig 3.** *Sabidius cornatus* (Holotype LACM-AHF POLY 655) (A). Anterior chaetigers, dorsal view; (B). Neuropodium with 2 curved modified spines and 1 with long arista.

lobed cuticle on anterior margin (Figs 2A–2C and 3A); median antennae, ciliated bands, terminal sensory organ, and cheek organs all absent; nuchal organs as curved slits on posterior margin (Fig 2A and 2C), with reddish–brown pigmentation, often fused to each other and extending across dorsum (Fig 3A). Eyes absent. Ventral mouth with saclike pharynx, everted in some specimens. Three prebranchial chaetigers (Figs 2A and 2B and 3A). Six to seven pairs of distinctly short, cirriform, dorsal branchiae; first four pairs less than 40 μm long, last three pairs slightly longer (Fig 2B); inner margin of branchiae basally ciliated, continuing through ciliary band across dorsum, connecting each pair (Fig 2D). First three chaetigers wider than long, following chaetigers as wide as long (Fig 2A and 2B). Parapodia biramous and papillated, both noto- and neuropodial lobes absent (Fig 2B), notochaetae emerging dorso-laterally, neurochaetae emerging laterally. Notochaetae all capillary, arranged in single row; prebranchial chaetigers with five notochaetae chaetae each; branchial and postbranchial chaetigers with four chaetae per fascicle (Fig 2A and 2B). Neurochaetae of three types: capillary chaetae,

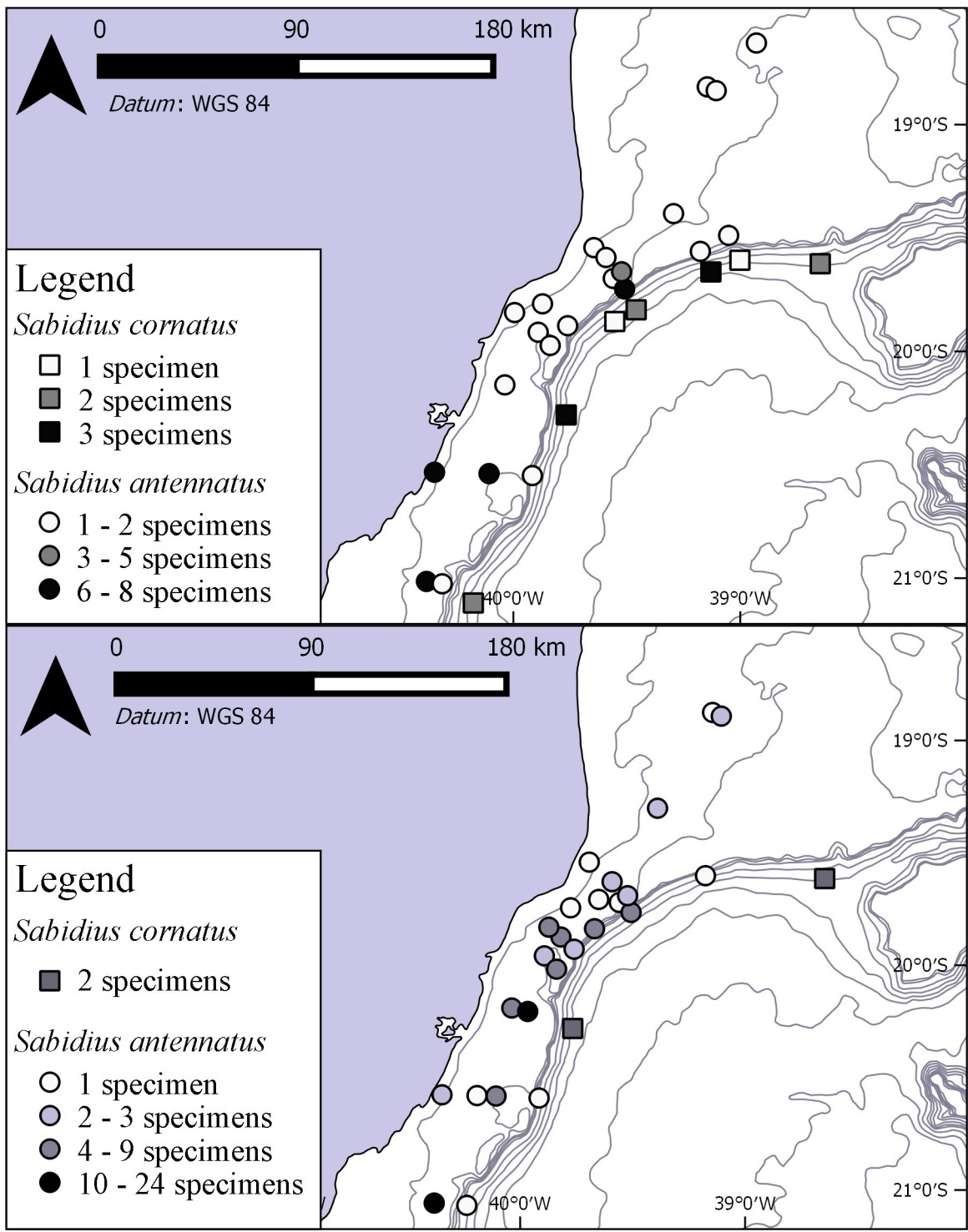

**Fig 4. Samplings of the species of *Sabidius* in summer (above) and winter (bellow).** Numbers in legend refers to abundance at each station.

curved modified spines without arista (Fig 2E) and curved modified spines with long arista (Fig 3B); neurochaetae always arranged in single rows; prebranchial chaetigers with five capillary neurochaetae each; branchial chaetigers with seven capillary neurochaetae each;

postbranchial chaetigers, until chaetiger 11, with four capillary chaetae per fascicle, from chaetiger 12 to end of body, neurochaetae as 5–6 enlarged, curved modified spines per fascicle (Fig 2E), of which at least 1–2 have long arista. Pygidium not observed, all specimens incomplete.

**Habitat.** Found in muddy sediments, with small amount of sand, 10.3–15.8˚C; between 140–3,388 m deep.

**Distribution.** Northern Atlantic Ocean: type locality off New England, from slope to abyssal depths, 400–2,946 m deep [4, 17], U.S Atlantic slope from off the Canadian boundary to the Carolinas [18]; slope depths off Louisiana, Gulf of Mexico [10]. Southern Atlantic Ocean: Southeastern Brazil, off Espírito Santo, 140–438 m (Fig 4). Pacific Ocean: off eastern side of Hokkaido Island, 3,388 m deep [2].

**Remarks.** The specimens herein examined match both the original description (Hartman, 1965) based on New England material, and the re-description, based on material from the Atlantic and Pacific Oceans [2]. Although both the original description and the re-description by Strelzov [2] did not mention the slight differences in branchial length along branchiate chaetigers, this character was confirmed when the type material was examined. Eyes were not visible neither in Brazilian specimens nor in the type material, although they were described as present in former descriptions [2, 4]. However, those are not true eyes but pigment spots instead [2], which were described as eyes by Hartman [4], while Strelzov [2] stated the eyes are not always conspicuous because the pigmentation readily fades after preservation. Branchial cilia were not mentioned either by Hartman [4] or Strelzov [2]. However, this character is only visible under SEM and this technique was not available by the time of those descriptions.

Among Brazilian material, specimens of *S. cornatus* were not found in habitats deeper than 438 m, although the survey which collected the material used for the present study sampled bottoms down to 3,000 m deep. This is the shallowest and the most restricted bathymetrically report of this species, within a range of 298 m only. Northern Atlantic material was obtained from slope and abyssal depths, within a bathymetric range of 2,500 m, and apparently those animals are more abundant in deeper bottoms [17].

***Sabidius antennatus* n. sp.** (Figs 4–6) urn:lsid:zoobank.org:act:4B4DBA02-0BB1-4D5D-B965-CAD474BC29E9.

**Material examined.** *Type series*. Holotype (MNRJP–1497): 77 chaetigers complete, 4.25 mm long, 0.12 mm wide, coll. 19 January 2012, 19˚43'14.34"S 39˚33'34.86"W, 45 m. Paratypes: 3 specs. (MNRJP–1492), coll. 15 Dec 2010, 19˚49'57.38"S 39˚52'14.02"W 29 m; 7 specs. (MNRJP–1496), coll. 15 Jan 2012, 19˚45'54.56"S 39˚30'25.23"W, 121 m; 4 specs. (MNRJP–1494), coll. 22 Jan 2012, 21˚3'27.14"S 40˚22'59.61"W, 36 m; 4 specs. (MNRJP–1495), coll. 21 Jan 2012, 20˚34'53.42"S 40˚06'27.43"W, 45 m; 1 spec. (MNRJP–1493), coll. 9 Dec 2011, 19˚31'51.66"S 39˚03'04.04"W, 140 m; 5 specs. (MZUSP–3869), coll. 12 Jul 2013, 20˚34'53.05"S 40˚06'27.68"W, 43 m; 24 specs. (MZUSP–3872), coll. 13 Jul 2013, 20˚12'21.46"S 39˚58'00.3"W, 45 m. All type specimens collected off Espírito Santo, southeastern Brazil.

**Additional material.** State of Espírito Santo: 2 specs. (MZUSP–3863), coll. 22 Jan 2012, 21˚04'01.29"S 40˚18'50.11"W, 46 m; 2 specs. (MNRJP–1499), coll. 21 Jan 2012, 20˚34'32.47"S 40˚20'52.37"W, 20 m; 1 spec. (MNRJP–1500), coll. 21 Jan 2012, 20˚35'25.16"S 39˚54'58.31"W, 145 m; 1 spec. (MNRJP–1501), coll. 20 Jan 2012, 20˚11'25.35"S 40˚02' 16.02"W, 35 m; 2 specs. (MNRJP–1502), coll. 19 Jan 2012, 19˚43'14.34"S 39˚33'34.86"W, 45 m; 1 spec. (MZUSP–3864), coll. 19 Jan 2012, 19˚26'05"S 39˚17'38.92"W, 46 m; 2 specs. (MNRJP–1498), coll. 15 Jan 2012, 19˚36'04.32"S 39˚10'34.07"W, 134 m; 1 spec. (MNRJP–1503), coll. 18 Jan 2012, 18˚52'32.61"S 39˚08'42.82"W, 34 m; 2 specs (MNRJP–1504), coll. 18 Jan 2012, 18˚53'29.72"S 39˚06'23.3"W, 43 m; 1 spec. (MNRJP–1485), coll. 17 Dec 2010, 19˚52'21.52"S 39˚59'33.54"W, 26 m; 1 spec. (MNRJP–1486), coll. 12 Dec 2010, 19˚35'12.39"S 39˚38'33.16"W, 29 m; 1 spec. (MNRJP–1487), coll. 16 Dec 2010, 19˚57'32.89"S 39˚53'30.69"W, 43 m; 1 spec. (MNRJP–

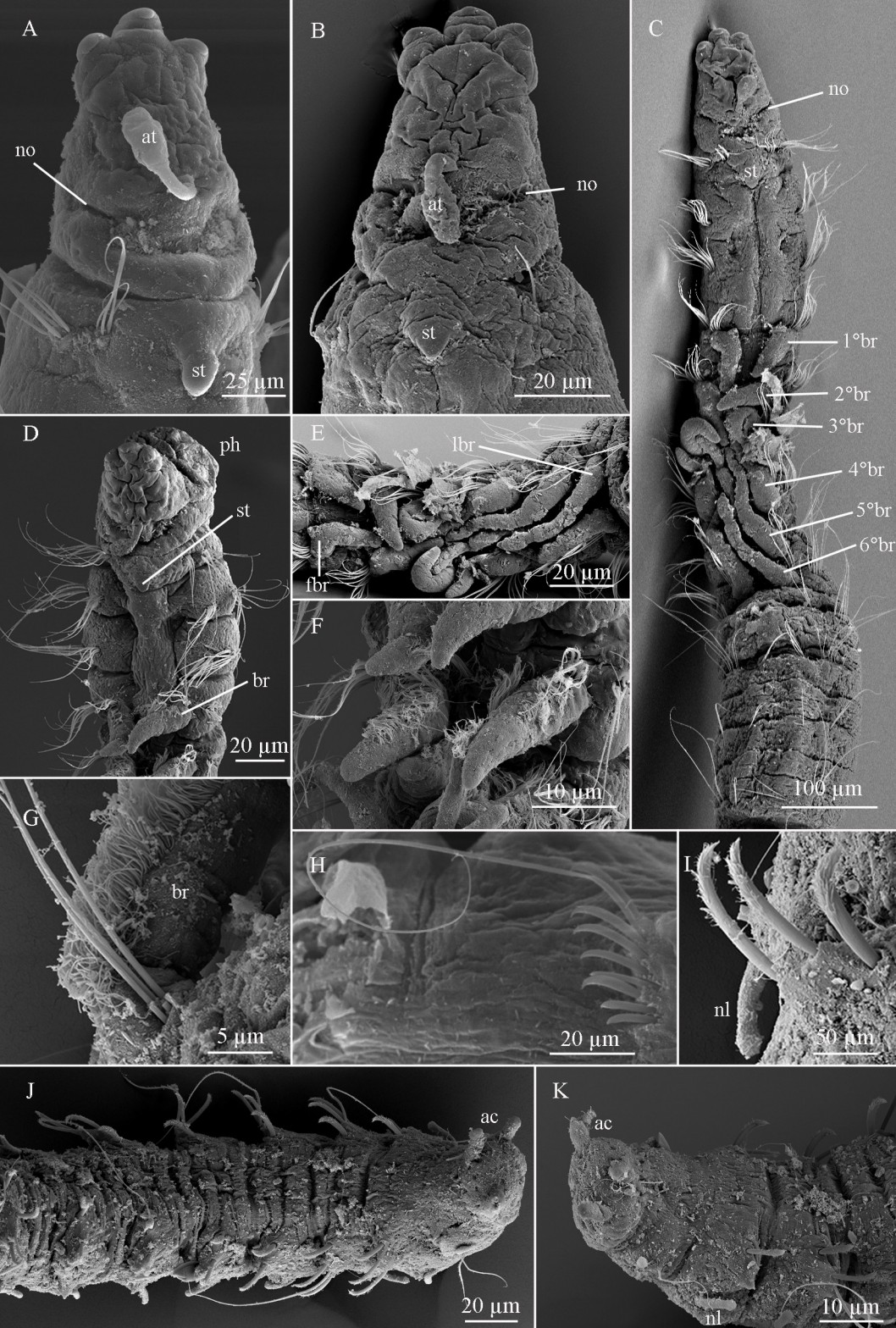

**Fig 5.** *Sabidius antennatus* sp. nov. (A–B). Prostomium, dorsal view; (C). Anterior chaetigers, dorsal view; (D). anterior chaetigers dorso-lateral view; (E). Branchial chaetigers, dorsal view; (F). Pair of branchiae, antero-lateral view; (G). Close up of one branchia, lateral view; (H). Neuropodium with 7 curved modified spines and 1 with long arista; (I). Curved modified spines; (J). Posterior chaetigers, ventral view; (K). Pygidium, ventral view. at = antenna; ac = anal cirri; br = branchiae; fbr = first branchiae; lbr = last branchiae; nl = notopodial lobe; no = nuchal organ; ph = everted pharynx; st = short tubercule.

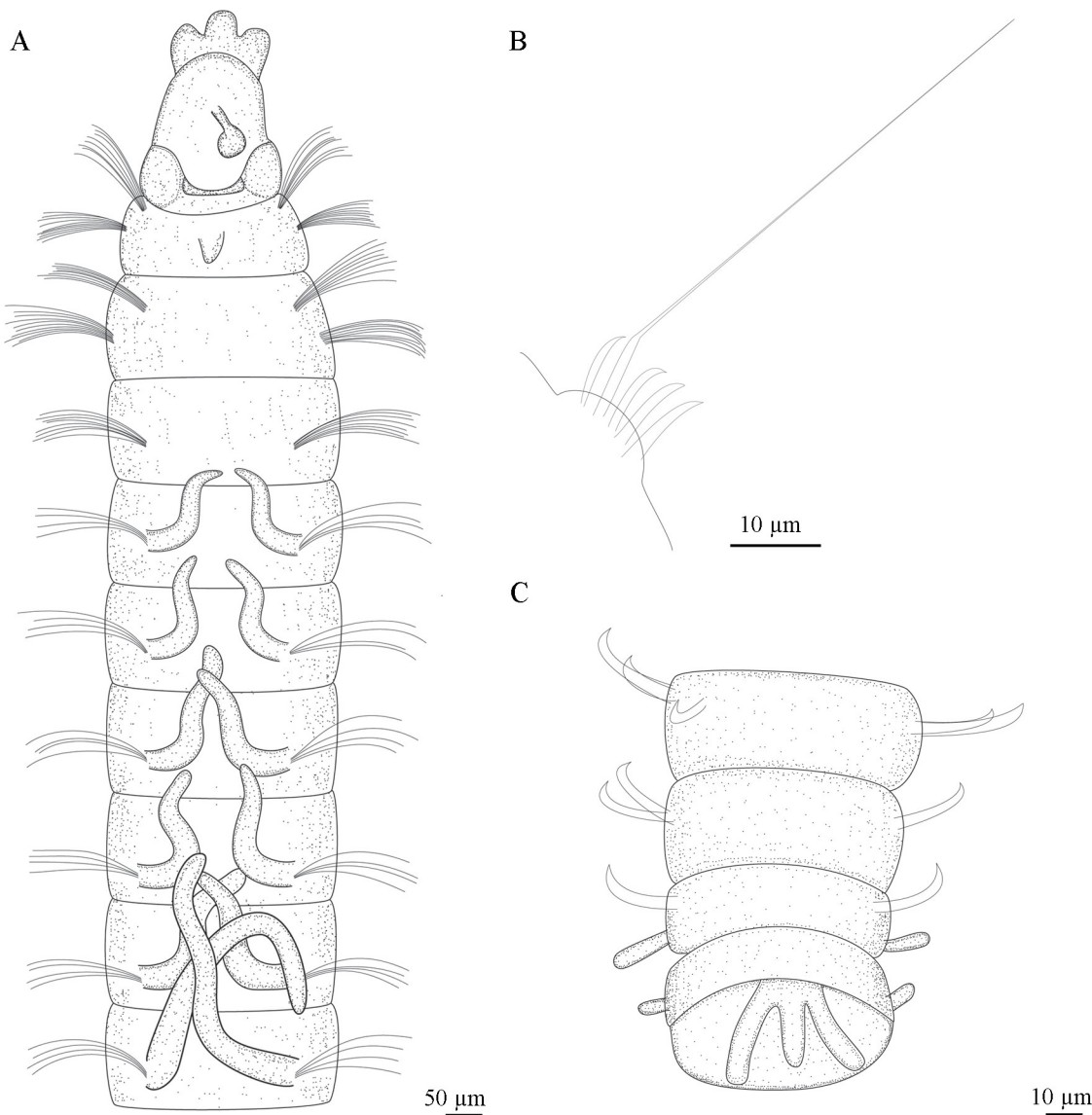

**Fig 6.** *Sabidius antennatus* n. sp. A). Anterior chaetigers, dorsal view; (B). Neuropodium with 5 curved modified spines and 1 with long arista; (C). Last chaetigers, ventral view.

1488), coll. 12 Dec 2010, 19˚37'48.27"S 39˚35'25.83"W, 38 m; 2 specs. (MNRJP–1489), coll. 16 Dec 2010, 20˚01'03.73"S 39˚50'13.76"W, 48 m; 1 spec. (MNRJP–1490), coll. 16 Dec 2010, 19˚55'44.66"S 39˚45'38.7"W, 46 m; 5 specs. (MNRJP–1491), coll. 12 Dec 2010, 19˚41'33.92"S 39˚31'17.74"W, 48 m; 3 specs. (MNRJP–1505), coll. 17 Jan 2012, 18˚40'55.3"S 38˚55'41.48"W, 44 m; 16 specs. (MZUSP–3865), coll. 11 Jul 2013, 21˚03'31.13"S 40˚22'59.88"W, 34 m; 1 spec. (MZUSP–3866), coll. 11 Jul 2013, 21˚04'04.56"S 40˚14'14.08"W, 147 m; 2 specs. (MZUSP–3867), coll. 18 Jan 2012, 18˚53'29.72"S 39˚6'23.3"W, 43 m; 1 spec. (MZUSP–3868), coll. 12 Jul 2013, 20˚34'47.13"S 40˚11'31.1"W, 34 m; 1 spec. (MZUSP–3870), coll. 13 Jul 2013, 20˚35'23.09"S 39˚55'01.18"W, 156 m; 5 specs. (MZUSP–3871), coll. 13 Jul 2013, 20˚11'25.75"S 40˚02'15.87"W, 33 m; 1 spec. (MZUSP–3873), coll. 14 Jul 2013, 19˚43'14.33"S 39˚33'34.78"W, 43 m; 6 specs. (MZUSP–3874); coll. 27 Jun 2013, 19˚45'53.43"S 39˚30'25.97"W, 138 m; 2 specs.

(MZUSP–3875), coll. 15 Jul 2013, 19˚18'06.12"S 39˚23'23.35"W, 33 m; 1 spec. (MZUSP–3876), coll. 29 Jun 2013, 19˚36'03.57"S 39˚10'33.64"W, 142 m; 1 spec. (MZUSP–3877), coll. 15 Jul 2013, 18˚52'31.35"S 39˚08'41.34"W, 33 m; 5 specs. (MZUSP–3878), coll. 16 Jul 2013, 18˚ 53'31.97"S 39˚06'21.78"W, 43 m; 3 specs. (MZUSP–3879), coll. 17 Jul 2011, 19˚57'32.36"S 39˚ 53'33.01"W, 46 m; 4 specs. (MZUSP–3880), coll. 16 Jul 2011, 19˚52'29.66"S 39˚49'08.1"W, 41 m; 1 spec. (MZUSP–3881), coll. 15 Jul 2011, 19˚42'26.81"S 39˚39'05.27"W, 36 m; 3 specs. (MZUSP–3882), coll.15 Jul 2011, 19˚37'41.83"S 39˚35'31.52"W, 35 m; 4 specs. (MZUSP– 3883), coll. 17 Jul 2011, 20˚01'02.6"S 39˚50'18.72"W, 49 m; 2 specs. (MZUSP–3884), coll. 16 Jul 2011, 19˚55'45.59"S 39˚45'41.35"W, 43 m; 5 specs. (MZUSP–3885), coll. 15 Jul 2011, 19˚ 50'16.39"S 39˚40'11.23"W, 46 m; 4 specs. (MZUSP–3886), coll. 13 Jul 2011, 19˚41'24.99"S 39˚ 31'20.42"W, 44 m; 1 spec. (MZUSP–3887), coll. 13 Jul 2011, 19˚32'28.16"S 39˚41'35.15"W, 11 m; 4 specs. (MZUSP–3888), coll. 16 Jul 2011, 19˚49'52.15"S 39˚52'24.51"W, 28 m; 1 spec. (MZUSP–3889), coll. 15 Jul 2011, 19˚44'44.06"S 39˚46'32.01"W, 29 m; 2 specs. (MZUSP– 3890), coll. 22 Jan 2012, 21˚03'27.14"S 40˚22'59.61"W, 36 m; 2 specs. (MZUSP–3891), coll. 21 Jan 2012, 20˚34'53.42"S 40˚06'27.43"W, 45 m; 2 specs. (MZUSP–3892), coll. 16 Dec 2010, 19˚ 57'32.89"S 39˚53'30.69"W, 43 m; 3 specs. (MZUSP–3893), coll.15 Jul 2011, 19˚37'41.83"S 39˚ 35'31.52"W, 35 m; 1 spec. (MZUSP–3894), coll. 15 Jan 2012, 19˚45'54.56"S 39˚30'25.23"W, 121 m; 1 spec. (MZUSP–3895), coll. 15 Jan 2012, 19˚36'04.32"S 39˚10'34.07"W, 134 m; 1 spec. (MZUSP–3896), coll. 21 Jan 2012, 20˚34'32.47"S 40˚20'52.37"W, 20 m; 1 spec. (MZUSP– 3897), coll. 18 Jan 2012, 18˚52'32.61"S 39˚08'42.82"W, 34 m.

**Diagnosis.** Prostomium with median antenna, nuchal organs as curved slits on posterior margin, associated with light brown pigmentation, often fused to each other and extending across dorsum. Short mid–dorsal tubercule on first chaetiger. Three prebranchial chaetigers; six pairs of well–developed dorso-lateral branchiae, one pair per chaetiger. Notopodial lobes only developed on posterior chaetigers; neuropodial lobes absent throughout. Pygidium with three short digitiform anal cirri.

**Description.** Complete specimen, with 77 chaetigers (50–75), 4.25 mm in length (4.47– 5.83 mm), 0.12 mm wide (0.09–0.16 mm). Body long and slender, fragile, easily broken. Pre-served specimens white–yellowish, without pigmentation patterns other than prostomial spots. Prostomium longer than wide, with distinctive thick, tri-lobed cuticle on anterior mar-gin; median antenna present, 0.05 mm (0.03–0.05 mm) in length, not reaching anterior margin of prostomium, digitiform with rounded expanded tip (Figs 5A–5D and 6A); ciliated bands, terminal sensory organ, and cheek organs all absent; nuchal organs as curved slits on posterior margin of prostomium, associated with light brown pigmentation, often fused to each other and extending across dorsum (Figs 5A, 5B and 6A). Eyes absent. Ventral mouth with saclike pharynx, everted in some specimens (Fig 5D). Three prebranchial chaetigers. Six pairs of well-developed, digitiform and distally tapered branchiae, progressively longer, last pairs twice as long as first pairs, or longer (Figs 5C, 5E–5G and 6A); 1$^{st}$ pair 45 μm long (40–60 μm), 2$^{nd}$ pair 55 μm long (50–70 μm), 3$^{rd}$ pair 70 μm long (60–80 μm), 4$^{th}$ pair 80 μm long (70–90 μm), 5$^{th}$ pair 105 μm long (80–120 μm), and 6$^{th}$ pair 115 mm long (90–140 mm); branchiae heavily cili-ated at bases, connected to each other across dorsum within pairs by ciliary bands (Fig 5C–5G). Chaetiger 1 wider than long, with short mid–dorsal tubercule (Figs 5A–5D and 6A) fol-lowing chaetigers almost as wide as long, last 10 chaetigers wider than long. Parapodia bira-mous and papillated, notochaetae emerging dorso-laterally, neurochaetae laterally, notopodial lobes present on last ~10 chaetigers (Fig 5J); neuropodial lobes absent throughout. Notochae-tae all capillary, arranged in single rows; prebranchial chaetigers with eight notochaetae each, branchial segments with four notochaetae per fascicle, postbranchial chaetigers with three. Neurochaetae always in single rows and of three types: capillary chaetae, curved modified spines without arista (Figs 5H–5J, 6B and 6C) and curved modified spines with long arista

(Figs 5H and 6B); prebranchial chaetigers with 10 capillary neurochaetae each; branchial chaetigers with seven capillary neurochaetae each; postbranchial chaetigers, until chaetiger 12, with six capillary chaetae, from chaetiger 13 to end, neurochaetae as 5–7 enlarged curved modified spines (Figs 5H, 5I, 6B and 6C), at least 1–2 of which per fascicle with long arista (Figs 5H, 5J and 6B). Pygidium with three digitiform anal cirri along ventral margin, all about same length (Figs 5K and 6C).

**Habitat.** Found in muddy-sand sediments, with few scattered boulders, 16.1–25.6°C; between 11–156 m deep.

**Distribution.** Southern Atlantic Ocean: southeastern Brazil, off Espírito Santo state, 11–156 m deep (Fig 4).

**Remarks.** Individuals of *S. antennatus* **n. sp.** are very similar morphologically to specimens of the other known species of this genus, *S. cornatus*. Animals of both species share the presence of a distinctive thick, tri-lobed cuticle on the anterior margin of the prostomium, similar types of modified chaetae, few pairs of branchiae, and the absence of neuropodial lobes. These two species, however, are differentiated because members of *S. antennatus* **n. sp.** have a median antenna and longer and more heavily ciliated branchiae.

Members of *S. antennatus* **n. sp.** also differ from individuals of *S. cornatus* in having a short mid–dorsal tubercule on chaetiger 1 and notopodial lobes on posteriormost chaetigers, but it is doubtful whether posteriormost chaetigers of members of *S. cornatus* were ever examined, as the specimens are very fragile and a complete specimen has never been registered.

The presence of a short mid-dorsal tubercule on chaetiger 1 is a unique characteristic among Paraonidae. The closest feature present within the family is a large mid-dorsal rounded papilla on first branchial segment of *Aricidea* (*Strelzovia*) *claudiae* Laubier, 1967 [24].

The pygidium of *S. cornatus* was not described neither in the original description of the species [4], nor in the re-description by Strelzov [2], due to the absence of complete specimens in both studies (as also herein). Blake [10] was the first to describe complete specimens and, according to the author, the pygidium has an extended smooth lobe, while in members of other genera of Paraonidae frequently there are 2–3 anal cirri. Due to the fragility of these animals, it is possible that Blake [10] described a damaged pygidium. In *S. antennatus* **n. sp.**, the pygidium has 3 digitiform anal cirri along the ventral margin, all of similar length.

Specimens described as *Aricidea* (*Acmira*) sp. C from northeastern Brazil, off Maranhão State, in a Ph.D. dissertation [21] match the description of *Sabidius antennatus* **n. sp.** These specimens were obtained from fine sand bottoms, 28–28.30°C, between 33–55 m depth. Although they were not examined, we strongly believe the specimens belong to *S. antennatus* **n. sp.**, due to the presence of antenna, same distribution of branchiae and size and shape of modified chaetae. Before expanding the distribution of *S. antennatus* **n. sp.** for the northeastern Brazil, however, those specimens should be examined.

In regard to the bathymetric distribution, individuals of *S. antennatus* inhabit shallower bottoms, at continental shelf depths.

**Etymology.** The specific name refers to the presence of a median antenna, which is herein described for the first time for members of *Sabidius*.

## Acknowledgments

We are thankful to CENPES/PETROBRAS for providing the material used for the present study. We are also grateful to Leslie Harris for all the support provided during a stay of NR in the NHMLA. We also thank Enio Mattos and Phillip Lenktaitis, from the Departamento de Zoologia, IB–USP, for preparing specimens for the SEM study and for operating the SEM equipment.

## Author Contributions

**Conceptualization:** Natália Ranauro, Rômulo Barroso, Paulo Cesar Paiva, João Miguel de Matos Nogueira.

**Data curation:** Natália Ranauro.

**Investigation:** Natália Ranauro.

**Resources:** Rômulo Barroso.

**Supervision:** Paulo Cesar Paiva, João Miguel de Matos Nogueira.

**Validation:** Rômulo Barroso.

**Visualization:** Rômulo Barroso, Paulo Cesar Paiva, João Miguel de Matos Nogueira.

**Writing – original draft:** Natália Ranauro.

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
