## [Decision Letter · Decision Letter 0]

29 Nov 2019

PONE-D-19-30964

On the genus Sabidius Strelzov, 1973 (Annelida: Paraonidae), with a redescription of the type species and the description of a new species.

PLOS ONE

Dear Dr. Ranauro,

Thank you for submitting your manuscript to PLOS ONE. After careful consideration, we feel that it has merit but does not fully meet PLOS ONE’s publication criteria as it currently stands. Therefore, we invite you to submit a revised version of the manuscript that addresses the points raised during the review process.

I agree with the referees about the quality and interest of this manuscript. I think that this work deserves to be published after few corrections. Reviewers have recommended some minor/major revisions to your manuscript. Therefore, I invite you to respond to the reviewers' comments and revise your manuscript. If the authors decide to do a manuscript revision for this journal, they need to carefully consider these comments particularly the the style of the description. English can be improved at some points as indicated by referee 1. Please, check the comments of the referees on the manuscript.

We would appreciate receiving your revised manuscript by 8th January 2020. To enhance the reproducibility of your results, we recommend that if applicable you deposit your laboratory protocols in protocols.io, where a protocol can be assigned its own identifier (DOI) such that it can be cited independently in the future. For instructions see: http://journals.plos.org/plosone/s/submission-guidelines#loc-laboratory-protocols

We look forward to receiving your revised manuscript.

Kind regards,

Marcos Rubal García, PhD

Academic Editor

PLOS ONE

Journal Requirements:

1. In your Methods section, please provide additional information regarding the permits you obtained for the work. Please ensure you have included the full name of the authority that approved the field site access and, if no permits were required, a brief statement explaining why.

2. In your Methods section, please provide additional location information of the sampling sites, including geographic coordinates for the data set if available.

3.

Please take this opportunity to be sure you have met all of our guidelines for new species. For proper registration of a new zoological taxon, we require two specific statements to be included in your manuscript.

In the Results section, the globally unique identifier (GUID), currently in the form of a Life Science Identifier (LSID), should be listed under the new species name, for example:

Anochetus boltoni Fisher sp. nov. urn:lsid:zoobank.org:act:B6C072CF-1CA6-40C7-8396-534E91EF7FBB

Another LSID for the manuscript itself should also appear within the Nomenclature statement. You will need to contact Zoobank (zoobank.org/About) to obtain a GUID (LSID). You should receive one LSID for your manuscript and a separate, unique LSID for the new species.

Please also insert the following text into the Methods section, in a sub-section to be called "Nomenclatural Acts":

*The electronic edition of this article conforms to the requirements of the amended International Code of Zoological Nomenclature, and hence the new names contained herein are available under that Code from the electronic edition of this article. This published work and the nomenclatural acts it contains have been registered in ZooBank, the online registration system for the ICZN. The ZooBank LSIDs (Life Science Identifiers) can be resolved and the associated information viewed through any standard web browser by appending the LSID to the prefix "" ext-link-type="uri" xlink:type="simple">http://zoobank.org/". The LSID for this publication is: urn:lsid:zoobank.org:pub: XXXXXXX. The electronic edition of this work was published in a journal with an ISSN, and has been archived and is available from the following digital repositories: PubMed Central, LOCKSS [author to insert any additional repositories]*.

All PLOS ONE articles are deposited in PubMed Central and LOCKSS. If your institute, or those of your co-authors, has its own repository, we recommend that you also deposit the published online article there and include the name in your article.

Following a recent ruling by the International Commission on Zoological Nomenclature, electronic journals are now a valid format for publication of new zoological taxa. In order to ensure the valid publication of your new species, please be sure to include the updated version of Nomenclatural Acts (above). A complete explanation of our guidelines for publishing new species can be found on our website: http://www.plosone.org/static/guidelines#zoological.

Reviewers' comments:

Reviewer's Responses to Questions

**Comments to the Author**

1. Is the manuscript technically sound, and do the data support the conclusions?

Reviewer #1: Yes

Reviewer #2: Yes

2. Has the statistical analysis been performed appropriately and rigorously? 

Reviewer #1: N/A

Reviewer #2: N/A

3. Have the authors made all data underlying the findings in their manuscript fully available?

Reviewer #1: Yes

Reviewer #2: Yes

4. Is the manuscript presented in an intelligible fashion and written in standard English?

Reviewer #1: No

Reviewer #2: Yes

5. Review Comments to the Author

Reviewer #1: The goals of the paper are plenty achieved with the material studied, the mthods applies and the results obtained; however, their importance is only of limted reach. Without considering whether it merits publication in this journal or in any other directly devoted to taxonomy, some flaws muts be fixed before acceptance. First of all, English language needs a thorough revisión in the abstract and discussion headings. Also figures need some reworking; wheras map are of good quality and SEM photogrphas are superb, ink drawings are of por quality and need to be worked out.

Reviewer #2: The review was made in the attached PDF document.

The manuscript provides a redescription of the type species of the polychaete genus Sabidius and description of a new species from Brazil. In general, it is well-written, the morphological characters used to designate the new species are well-defined and appropriate considering the current knowledge on paraonids, and the species are illustrated with both line drawings and SEM images. However, I have provided suggestions to improve the reading and the descriptions. One major edition that the authors should follow is the style of the description. It is common practice in taxonomy that the descriptions are based on the holotypes and the variation should be reported for the paratypes (either within parentheses or within the text).

6. PLOS authors have the option to publish the peer review history of their article (what does this mean?). If published, this will include your full peer review and any attached files.

Reviewer #1: No

Reviewer #2: No

---

## [Author Response · Author response to Decision Letter 0]

4 Feb 2020

Reviewer 1: The goals of the paper are plenty achieved with the material studied, the methods applies and the results obtained; however, their importance is only of limited reach. Without considering whether it merits publication in this journal or in any other directly devoted to taxonomy, some flaws must be fixed before acceptance. First of all, English language needs a thorough revision in the abstract and discussion headings. Also figures need some reworking; whereas map are of good quality and SEM photographs are superb, ink drawings are of poor quality and need to be worked out.

NR: One of the main goals in zoology is to understand the relationships between different metazoan taxa. Researchers estimate that only half of the total diversity of annelids has been described. Only describing new species we can fully understand the phylogenetic relationships within each phylum. Describing new species and new morphological features can help to create new hypothesis of phylogenetic relationships within the family Paraonidae and in the Annelida phylogeny. For the family Paraonidae, most of the specimens available in the museums were fixed with formalin, what makes hard gene extraction for molecular analysis. This highlights the importance of morphological research and description of new species. English language has been reviewed. The drawings were modified, although they loose quality when converted into pdf format.

Reviewer 2: The manuscript provides a redescription of the type species of the polychaete genus Sabidius and description of a new species from Brazil. In general, it is well-written, the morphological characters used to designate the new species are well-defined and appropriate considering the current knowledge on paraonids, and the species are illustrated with both line drawings and SEM images. However, I have provided suggestions to improve the reading and the descriptions. One major edition that the authors should follow is the style of the description. It is common practice in taxonomy that the descriptions are based on the holotypes and the variation should be reported for the paratypes (either within parentheses or within the text).

NR: The style of the description was modified as suggested.

---

## [Editor Report · Decision Letter 1]

13 Feb 2020

On the genus Sabidius Strelzov, 1973 (Annelida: Paraonidae), with a redescription of the type species and the description of a new species.

PONE-D-19-30964R1

Dear Dr. Ranauro,

We are pleased to inform you that your manuscript has been judged scientifically suitable for publication and will be formally accepted for publication once it complies with all outstanding technical requirements.

With kind regards,

Marcos Rubal García, PhD

Academic Editor

PLOS ONE

Additional Editor Comments (optional):

Authors have done all the changes suggested by the referees, and the manuscript is now suitable for publication.
---

## [Editor Report · Acceptance letter]

18 Feb 2020

PONE-D-19-30964R1 

On the genus *Sabidius* Strelzov, 1973 (Annelida: Paraonidae), with a redescription of the type species and the description of a new species. 

Dear Dr. Ranauro:

I am pleased to inform you that your manuscript has been deemed suitable for publication in PLOS ONE. Congratulations! Your manuscript is now with our production department. 

With kind regards,

on behalf of

Dr. Marcos Rubal García 

Academic Editor

PLOS ONE